# Plasma Somatostatin Levels Increase during Scoliosis Surgery, but Not Herniated Disc Operations: Results of a Pilot Study

**DOI:** 10.3390/biomedicines11082154

**Published:** 2023-07-31

**Authors:** Balázs Sütő, Bálint Kolumbán, Éva Szabó, Sára Pásztor, Timea Németh, Teréz Bagoly, Bálint Botz, Erika Pintér, Zsuzsanna Helyes

**Affiliations:** 1Department of Anaesthesia and Intensive Therapy, Medical School, University of Pécs, 7624 Pécs, Hungary; suto.balazs@pte.hu; 2Department of Neurosurgery, Medical School, University of Pécs, 7623 Pécs, Hungary; 3Department of Otorhinolaryngology, Medical School, University of Pécs, 7621 Pécs, Hungary; 4Department of Internal Medicine, Fejér County Szent György University Teaching Hospital, 8000 Székesfehérvár, Hungary; 5Department of Languages for Biomedical Purposes and Communication, Medical School, University of Pécs, 7624 Pécs, Hungary; 6Department of Pharmacology and Pharmacotherapy & Eötvös Loránd Research Network, Chronic Pain Research Group, Medical School, University of Pécs, 7624 Pécs, Hungary; 7Department of Medical Imaging, Medical School, University of Pécs, 7624 Pécs, Hungary; 8National Laboratory for Drug Research and Development, 1117 Budapest, Hungary

**Keywords:** neuropeptides, pain, inflammation, tissue damage, scoliosis and disc herniation, Cobb angle, microdiscectomy, orthopedic and neuro-spine surgery, radioimmunoassay

## Abstract

Somatostatin (SST) released from capsaicin-sensitive sensory nerves in response to stimulation exerts systemic anti-inflammatory, analgesic actions. Its elevation correlates with the extent of tissue injury. We measured plasma SST alterations during spine operations (scoliosis and herniated disc) to determine whether its release might be a general protective mechanism during painful conditions. Sampling timepoints were baseline (1), after: soft tissue retraction (2), osteotomy (3), skin closure (4), the following morning (5). Plasma SST-like immunoreactivity (SST-LI) determined by radioimmunoassay was correlated with pain intensity and the correction angle (Cobb angle). In scoliosis surgery, postoperative pain intensity (VAS 2.) 1 day after surgery significantly increased (from 1.44 SEM ± 0.68 to 6.77 SEM ± 0.82, *p* = 0.0028) and positively correlated with the Cobb angle (*p* = 0.0235). The baseline Cobb degree negatively correlated (*p* = 0.0459) with the preoperative SST-LI. The plasma SST-LI significantly increased in fraction 3 compared to the baseline (*p* < 0.05), and significantly decreased thereafter (*p* < 0.001). In contrast, in herniated disc operations no SST-LI changes were observed in either group. The VAS decreased after surgery both in the traditional (mean 6.83 to 2.29, *p* = 0.0005) and microdiscectomy groups (mean 7.22 to 2.11, *p* = 0.0009). More extensive and destructive scoliosis surgery might cause greater tissue damage with greater pain (inflammation), which results in a significant SST release into the plasma from the sensory nerves. SST is suggested to be involved in an endogenous postoperative analgesic (anti-inflammatory) mechanism.

## 1. Introduction

Sensory neuropeptides released from the capsaicin-sensitive peptidergic sensory nerves play crucial roles in regulating various physiological functions, including tissue homeostasis, immune responses, endocrine signalling, and neural processes associated with pain and inflammation. These sensory fibres do not only transmit sensory input and pain signals to the central nervous system (afferent functions), but they also have important local and systemic efferent actions [1,2,3,4,5,6]. Understanding the role of these neuropeptides can provide valuable insights into the mechanisms underlying nociception and inflammation, paving the way for potential therapeutic interventions.

Proinflammatory neuropeptides, such as tachykinins (substance P, neurokinin A) and calcitonin gene-related peptide (CGRP) mediate vasodilation, plasma protein extravasation, and immune cell activation in the innervated area collectively called neurogenic inflammation [2,3]. Moreover, inhibitory peptides, with somatostatin (SST) being the most prominent among them, are released from the same nerve terminals in response to inflammation and tissue damage. The actions of SST extend beyond its local effects, it reaches remote sites through the systemic circulation, and suggests a therapeutic potential to modulate inflammation and pain [3,4,5,7]. SST acts as a modulator of inflammation by inhibiting the release of proinflammatory cytokines, chemokines, neuropeptides, and other mediators. It suppresses the activity of immune cells and reduces the recruitment and activation of inflammatory cells, thus attenuating the inflammatory response. SST helps to limit tissue damage and promote the resolution of inflammation. All these divergent protective actions of SST are mediated through interactions with five Gi-protein-coupled receptors (sst_1_–sst_5_) [8,9,10]. Among these receptors, sst_1_ and sst_4_ have been identified as key targets in mediating the anti-inflammatory and analgesic effects of SST [11]. These receptors are located on various immune cells, vascular endothelial cells, and nerve terminals. When SST binds to these receptors, it triggers intracellular signalling pathways that inhibit adenylate cyclase and the production of cAMP, which results in a consequent inflammatory mediator release.

We have shown earlier both in animal models and humans that SST is released from the capsaicin-sensitive afferents in response to inflammatory processes (arthritis, sepsis) and tissue damage (e.g., surgery) [12,13]. Data obtained in experiments conducted on rats, mice, and guinea pigs demonstrated that the released SST does not only interact with inflammatory cells locally, but also enters the systemic circulation. This enables it to exert anti-inflammatory and antinociceptive “sensocrine” effects in distant areas of the body [14,15]. Selective electrical or capsaicin-induced chemical activation of the peptidergic nociceptive fibres in rats inhibits the cardiorespiratory reflex responses (elevation of blood pressure, heart rate, and respiratory rate) [16]. In the adjuvant-induced chronic arthritis rat model, SST is released from the capsaicin-sensitive sensory nerves, inducing a four-fold elevation in its plasma concentration by the end of the 21-day examination period, which inhibits oedema formation and pain behaviour [11]. Exogenous SST and its synthetic analogues reduce both neurogenic and non-neurogenic inflammatory and nociceptive mechanisms [17,18,19,20], as well as inhibit the severity and increase the survival rate of experimental sepsis [21,22,23]. In addition to all these extensive animal experimental investigations, we demonstrated in humans that the plasma SST level increases during and after abdominal [24], thoracic, and some orthopaedic surgical interventions, as well as in sepsis [12,13,25]. These data might suggest a general endogenous protective mechanism mediated by SST released from the capsaicin-sensitive nerves.

In light of the dense innervation of the spinal column and intervertebral joints by peptidergic nociceptive fibres, our study aimed to assess changes in plasma SST-like immunoreactivity (SST-LI) during scoliosis orthopaedic surgery and disc hernia neurosurgical interventions, both conducted under general anaesthesia. During these operations, we monitored the levels of SST-LI in the bloodstream to understand the dynamics of SST release in response to the surgical procedure and pain. The objectives were to establish the correlation between the plasma SST levels and the intensity of pain, as well as the extent of the tissue damage incurred during the procedures and to investigate whether the release of SST could potentially serve as a general protective mechanism during painful conditions and surgical procedures.

## 2. Materials and Methods

### 2.1. Patients

Altogether 30 spine surgery patients undergoing orthopaedic and neurosurgical interventions were enrolled in our study, which lasted for 8 weeks. Patient numbers in the different groups and their mean ages are summarized in Table 1. Written informed consents were obtained prior to participation in all cases. The study established exclusion criteria that encompassed several factors, including patients who were below the age of 18, individuals suffering of any forms of autoimmune disease, pregnant women, and those who did not provide informed consent for participation in the research. The study was conducted in accordance with the Declaration of Helsinki, the protocol was approved by the Ethics Committee of the University of Pécs with the permission number 3362/5636-PTE-2017.

### 2.2. Surgical Treatment for Scoliosis, Disc Hernia, and Visual Analog Scale (VAS)

Orthopaedic surgery offers a treatment approach for scoliosis through the utilization of segmental fixation of the spine with Cotrel–Dubousset instrumentation. This technique provides patients with the ability to swiftly return to their normal lives and activities following the procedure. To assess the severity of spinal deformities in scoliosis, the Cobb angle is commonly employed. This measurement involves determining the sum of the tilt angles of the upper and lower end vertebrae, providing a quantitative evaluation of the magnitude of the deformity.

In neurosurgery, disc hernia operations aim to alleviate nerve compression by removing small fragments of the intervertebral disc, bone, and ligaments. The primary goal of these procedures is to free the affected nerve root by eliminating any disc fragments and degenerated disc material that may be causing compression. Traditional discectomy techniques typically involve larger incisions and significant muscle retraction. However, the advent of microdiscectomy has revolutionized the treatment of herniated discs, offering a minimally invasive alternative. This procedure requires only a small incision and employs surgical glasses or a microscope to magnify the operative site, facilitating the meticulous manipulation of the nerve root. By utilizing microsurgical tools, the surgeon can operate within the confined space of the spine. This approach allows for the precise removal of the disc material causing the compression while minimizing trauma to the surrounding tissues.

We assessed pain intensity using a standardized visual analogue scale (VAS). VAS is a commonly used tool for clinicians to assess the pain intensity of patients after surgery. VAS is a simple method to use, it is a straight line with two endpoints: it starts from a low intensity value (no pain at all: 0) and ends with an extreme level of pain (unbearable pain sensation: 10). There is a numerical scale between the two endpoints, and the patient has to assess a value according to the pain sensation. A higher measurable value illustrates higher pain intensity, while a lower value points out less pain sensation.

### 2.3. Medication

Prior to the surgical procedure, premedication was administered to the patients, involving the use of midazolam torrex (0.07 mg/kg body weight; Chiesi Pharmaceuticals GmbH, Vienna, Austria) and atropine (0.01 mg/kg; EGIS Gyógyszergyár Zrt., Budapest, Hungary). The induction of general anaesthesia was initiated through the intravenous administration of 1% propofol (1.5–2.5 mg/kg; Fresenius Kabi Deutschland GmbH, Bad Homburg, Germany) and fentanyl (0.0015 mg/kg; Richter Gedeon Nyrt., Budapest, Hungary). Following intubation, the patients were placed on mechanical ventilation to maintain adequate oxygenation. A balanced inhalation anaesthesia approach was employed, utilizing a mixture of oxygen and medical air in a 1:2 volume ratio, along with sevoflurane at a concentration of 1.6–2% (*v*/*v*; Abbott Laboratories, Wiesbaden, Germany). During the surgical procedure, the major opioid analgesic fentanyl was administered intravenously at doses ranging from 25 to 50 μg according to the patient’s physiological parameters.

Muscle relaxation was achieved using atracurium (0.5 mg/kg; GlaxoSmithKline Pharmaceuticals S.A., Poznań, Poland) during induction, and a maintenance dose of 10 mg every 20 min was administered intravenously. Towards the end of the operation, the effect of the muscle relaxant was reversed by administering a combination of neostigmine and atropine intravenously (2.5 mg/0.5 mg; Pharmamagist Kft., Budapest, Hungary and EGIS Gyógyszergyár Zrt., Budapest, Hungary).

For effective postoperative pain management, intravenous morphine (1%; 1–2 mg; TEVA, Hungary) was administered as needed. Furthermore, intravenous doses of Tramadol (50–100 mg; Grünenthal GmbH, Aachen, Germany), Diclofenac (50–75 mg; Novartis, Basel, Switzerland), and Paracetamol (1 g; Fresenius Kabi Hungary Kft., Budapest, Hungary) were given according to pharmaceutical instructions to address pain as required after the operation.

### 2.4. Blood Sampling

The procedures were conducted under the administration of general anaesthesia, which was augmented with the use of parenteral opioids for pain management. Prior to the surgical procedures, patients adhered to our hospital’s established surgical protocols, which involved fasting in accordance with the “nil by mouth” guideline starting from midnight on the day preceding the scheduled surgery. In both groups of orthopaedic and neurosurgical patients, blood samples were collected on five separate occasions to measure the levels of SST-like immunoreactivity (SST-LI) in the plasma. The first sample was obtained at the outset, before any surgical intervention took place (sample 1). Subsequently, additional samples were taken after the retraction of muscles, ligaments, and soft tissues (sample 2), following osteotomy or herniotomy (sample 3), at the time of closing the skin (sample 4), and the next morning at 8 o’clock (sample 5). To prevent clotting, two separate 5 mL blood samples were immediately collected in Vacutainers containing EDTA (18 mg REF 367525 and 143 I.U. REF 367674) that were maintained at a low, ice-cold temperature. Furthermore, to inhibit enzymatic degradation, 200 µL of the peptidase inhibitor aprotinin (Trasylol, Bayer Health-Care, Leverkusen, Germany) was promptly added to the sample intended for somatostatin measurement. Following centrifugation at 1000 rpm for 5 min and subsequently at 4000 rpm for 10 min, the plasma was frozen and stored at −70 °C until further analysis. All samples were analysed under standardized conditions at the end of the study.

It should be noted that previous investigations have indicated that a precise determination of SST-LI using the radioimmunoassay (RIA) requires 10 mL of blood and the subsequent acquisition of 6 mL of plasma. In cases where smaller volumes are used, the radioactivity measurements may not reliably align with the standard curve of the assay.

### 2.5. Determining Plasma SST-LI by RIA

The RIA method has been broadly used as a laboratory technique in clinical practice and research to precisely measure the concentration of different peptides and proteins. Its specificity and sensitivity allow this method to determine low concentrations of substances in biological samples (e.g., in the blood plasma). A highly specific antibody binds to the molecule, a radioactive tracer (labelled radioactive isotope) helps to measure the formed antigen-antibody complexes. After a certain incubation time, the antibodies compete with the radioactive antigen for the binding sites. Unbound antigens have to be removed from the complexes and the radioactivity of the separated complexes is measured. The higher the concentration of the substances, the lower the quantity of the radioactive labelled antigen that will be bound to the specific antibodies. After a calibration method the actual concentration of the substance can be measured. Plasma SST-LI was determined with a specific and sensitive RIA technique developed and validated by us as described earlier in detail [12,13,26]. We employed a C-terminal sensitive antiserum specific to SST-14, which demonstrated its ability to bind both biologically active forms of SST containing 14 and 28 amino acids. To extract the peptide from plasma, we utilized absolute alcohol in a ratio of 3:1. Following precipitation and centrifugation (2000 rpm for 10 min at 4 °C), the samples were dried using a nitrogen flow. Prior to RIA determination, the dried samples were reconstituted in an assay buffer. Our extraction and sample preparation technique exhibited a recovery rate of 79.8%.

### 2.6. Statistical Analysis

Statistical analysis was performed using the GraphPad Prism 6.0 (GraphPad Software, San Diego, CA, USA) software. Results are expressed as means ± standard errors of the mean (SEM). Comparisons between the SST levels measured at different time points within the same group were performed by repeated measures one-way ANOVA, followed by Tukey’s multiple comparison test. The VAS changes within groups were analysed by paired samples *t*-test. Correlations between the plasma SST levels and other laboratory parameters were assessed by linear regression and Pearson correlation. Plasma SST-LI data were tested for outlier values using the ROUT method in all groups in an identical manner. For all statistical analyses *p* < 0.05 was accepted as significant.

## 3. Results

### 3.1. Pain Intensity Significantly Increases and Positively Correlates with the Cobb Angle after Scoliosis Surgery

We made an assessment individually for each patient regarding the initial pain intensity (VAS 1.) and the level of pain after scoliosis surgery on day 1 (VAS 2.) using the VAS method. VAS 2. significantly increased in patients after scoliosis surgery (Figure 1A) on day 1 to the initial values. In patients undergoing scoliosis surgery, postoperative pain intensity (VAS 2.) positively correlated with the level of surgical correction (Cobb angle) among the patients (Figure 1B). The higher the Cobb angle (level of correction), the greater the postoperative pain intensity.

### 3.2. Plasma SST-LI Significantly Increases and Negatively Correlates with the Initial Cobb Angle in Scoliosis Surgery

As we measured the level of plasma SST-LI activity during the operation on five different occasions, the plasma SST-LI activity significantly increased in fraction 3 compared to fraction 1. Plasma SST-LI activity peaked in fraction 4, followed by a significant drop in fraction 5, (Figure 2A). Plasma SST-LI negatively correlated to the initial Cobb angle (Cobb1: deformity of the spinal column). The higher the level of deformity (Cobb1 angle), the lower the plasma SST-LI (Figure 2B).

### 3.3. VAS 2. Significantly Decreases in Both Traditional and Microdiscectomy Groups after Disc Hernia Surgery

In disc hernia operations, initial pain intensity (VAS 1.) was high in both the traditional and microdiscectomy surgical groups, which is basically attributed to the nature of the original disease in these patients. Pain after the operation (VAS 2.) significantly and similarly decreased after both types of interventions (Figure 3A,B).

### 3.4. Plasma SST-LI Does Not Change Significantly after Traditional or after Microdiscectomy Type of Disc Hernia Operations

Plasma SST-LI did not alter in disc herniated patients regardless of the type of the operation (Figure 4A,B). No significant change of SST-LI was detected in either group at any of the different sampling time points. There was a slight and steady increasing tendency of SST-LI in the microsdiscectomy group, which did not reach the level of statistical significance.

## 4. Discussion

The present pilot clinical study provides the first findings for the significant increase in plasma SST-LI during the orthopaedic scoliosis operation. However, a remarkable change in the plasma SST-LI was not detected after either a traditional or microdiscectomy type of disc hernia neurosurgical intervention performed under general anaesthesia. It is well-established that SST released from the peptidergic nociceptive fibers exhibits analgesic and anti-inflammatory effects [3,4,5,7,27,28]. Therefore, we hypothesized that its release might be an adaptive mechanism aimed at alleviating pain and reducing tissue damage. These observations are in agreement with earlier human data findings in other types of abdominal, thoracic, and orthopaedic surgeries and systemic inflammation (sepsis) demonstrating plasma SST-LI elevations [3,12,13,24]. The studies have reported findings indicating a noticeable yet statistically significant increase of approximately 10% in SST-LI within the systemic circulation following laparoscopic cholecystectomy, inguinal hernia repair, and abdominal wall hernia repairs [24]. Notably, we demonstrated earlier that the elevation of SST-LI appears to be even more prominent in cases of thoracic and orthopaedic surgeries characterized by more substantial tissue damage [12,13].

We suggest that SST is released from the activated peptidergic capsaicin-sensitive fibres in response to tissue injury and/or inflammatory mediators and enters into the bloodstream, inducing systemic analgesic/anti-inflammatory effects. Although we have no direct evidence for this theory by our descriptive results, a broad range of earlier animal experimental data in inflammation and pain models might provide a sufficient explanation [3,4,5,6,7,11,29]. However, besides the activated sensory nerves, it is important to note that SST can also be released from the inflammatory cells within the site of surgery. Since the blood samples were collected from patients who had undergone a 12-h fasting period, it is highly unlikely that gastrointestinal SST release contributed to the elevated levels [30].

During scoliosis surgery, plasma SST-LI significantly increased, presumably in response to tissue damage throughout the whole operative procedure, but no increase was observed in disc hernia operations. Scoliosis surgery caused more extensive tissue damage and remarkable pain, as shown by the significantly increased VAS 2. values during the first postoperative day. The greater the correction degree of the spinal column (Cobb angle), the higher the pain intensity, which is explained by the more extensive and stressful surgical procedure resulting in greater damage on skeletal muscles, joints, and tendons. This triggers inflammatory processes with consequent sensory nerve activation and SST release into the circulation. The neurosurgical interventions to remove the ruptured and herniated intervertebral discs represent remarkably smaller tissue damage and sensory nerve stimulation [31]. Furthermore, in disc hernia patients, the initial pain score is higher, but it is reduced after the operation when the etiological factor is eliminated.

The plasma SST-LI level negatively correlates with the initial Cobb angle (Cobb 1): the greater the deformity of the spinal column, the lower the plasma SST-LI level of the patients. SST might be depleted from the sensory nerves in patients with a greater deformity of the spinal column in response to continuous activation and ongoing chronic pain. Moreover, scoliosis is likely to be due to genetic defects and environmental factors [32], which might lead to a dysfunction of the sensory and sympathetic nervous systems [33].

SST inhibits activity-dependent nociceptors, and reduces pain sensation and perception following intrathecal and epidural administration [34], therefore, it was suggested as an alternative for opioid analgesics after minor surgical procedures, as well as in arthritis and neuropathy [27,34,35].

The role of SST in pain and inflammation is complex, involving multiple signalling molecules and pathways in neuronal and immune cells. Identifying the specific role and effects of SST within the intricate network can be very challenging. When tissue damage or an inflammatory stimulus occurs, immune cells, such as macrophages and lymphocytes, are activated and release various signalling molecules including cytokines and chemokines, as well as further neuropeptides. These molecules modulate pain and inflammation by acting on immune cells at the site of injury/surgical site. Moreover, SST reduces the release of inflammatory mediators such as cytokines and histamine from immune cells [36,37], and modulates the activity of T cells and macrophages, which are involved in the inflammatory responses [38,39,40]. The local administration of SST in animal studies significantly reduced pain behaviours, which can potentially be attributed to beta-endorphin release [28]. Based on the findings of this study, it can be inferred that the release of SST into the systemic circulation, most likely originating from the sensory nerves, appears to function as a widespread protective mechanism during the operative procedures. These interventions encompass scoliosis orthopaedic surgery and disc hernia neurosurgical interventions, which are characterized by substantial tissue damage, intense pain, and the presence of inflammatory responses. SST release during these interventions suggests its potential role in mitigating the detrimental effects, thereby highlighting its significance as a potential protective agent against tissue injury, pain, and inflammation in the context of surgery.

A limitation of the present research is the relatively small sample size involved in the study, which might not be well representative and conclusive for a larger population. Another limitation is that it did not thoroughly address the specific types of pain medication, including various analgesic drugs that patients had been using prior to the surgery. Therefore, it leaves a gap in the understanding and analysing the potential influence or interactions between pre-existing medication regimens and postoperative pain management strategies. Nevertheless, in the future, we aim to increase the number of patients and carry out further investigations including pharmacotherapy-based subgroup analysis.

In our study we could establish a link between SST levels and the intensity of patients’ pain, as well as the extent of tissue damage resulting from medical procedures. Further data could help to understand the multifaceted role of SST in inflammation and pave the way for its potential clinical implementation as a valuable intervention in the field of postoperative pain management.

## 5. Conclusions

SST release into the systemic circulation presumably from the sensory nerves is likely to be a general protective mechanism during spinal surgical procedures: scoliosis orthopedic surgery and disc hernia neurosurgical interventions with extensive tissue injury, pain, and inflammation. Tissue damage and pain sensation might increase plasma SST levels in patients during surgery.

The results of this study have the potential to enhance our understanding of the neurophysiological mechanisms underlying pain and tissue response during spinal operations. By elucidating the role of SST, we can potentially explore novel therapeutic approaches that target the modulation of neuropeptide signaling to optimize pain management and improve patient outcomes in the context of surgical interventions.

## Figures and Tables

**Figure 1 biomedicines-11-02154-f001:**
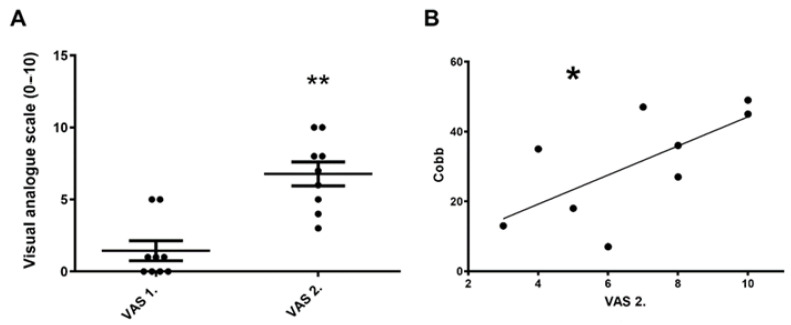
(**A**): Postoperative pain intensity increasing after scoliosis surgery on day 1. VAS 1. represents the initial pain and VAS 2. refers to the postoperative day 1 (paired samples *t*-test, mean ± SEM, ** *p* < 0.01). (**B**): Postoperative pain intensity positively correlates with the Cobb angle in scoliosis patients (Cobb = Cobb2 − Cobb1). (VAS = visual analogue scale, Cobb = quantifies the magnitude of deformity, parametric (Pearson) correlation and regression analysis * *p* < 0.05).

**Figure 2 biomedicines-11-02154-f002:**
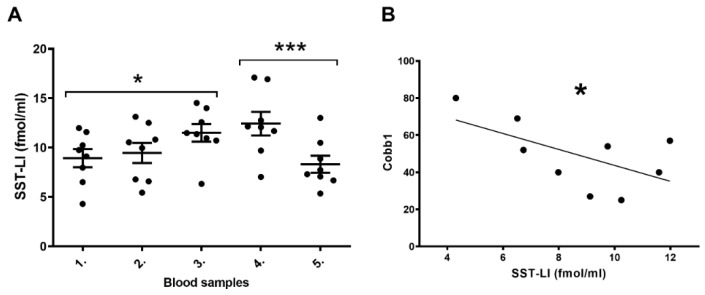
(**A**): Plasma SST-LI increases during scoliosis operation (repeated measures ANOVA and Tukey’s multiple comparison test, * *p* < 0.05, *** *p* < 0.001, means ± standard errors of mean). (**B**): Initial Cobb degree negatively correlates with the plasma SST-LI (parametric (Pearson) correlation and regression analysis * *p* < 0.05).

**Figure 3 biomedicines-11-02154-f003:**
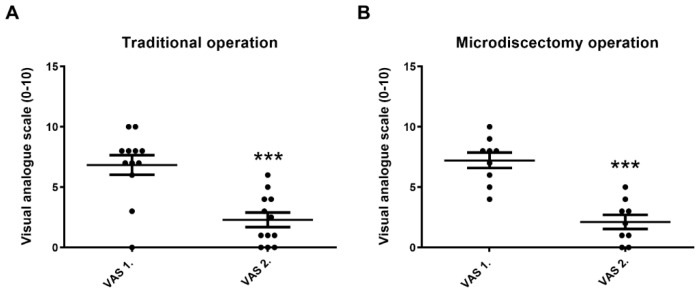
(**A**,**B**): Pain intensity decreases after disc hernia operations (parametric (Pearson) correlation and regression analysis, means ± SEM, *** *p* < 0.001).

**Figure 4 biomedicines-11-02154-f004:**
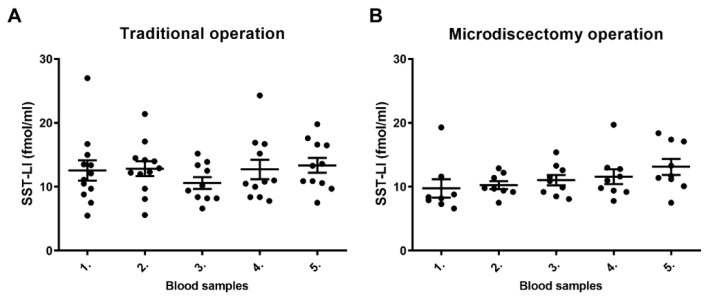
(**A**,**B**): Plasma SST-LI does not alter significantly during and after traditional or microdiscectomy type of disc hernia operations (parametric (Pearson) correlation and regression analysis means ± SEM).

**Table 1 biomedicines-11-02154-t001:** Number of patients and mean age in the scoliosis and disc hernia groups.

	Scoliosis Patients	Disc Hernia Patients (Traditional Group)	Disc Hernia Patients (Microdiscectomy Group)
Males	2	8	5
Females	7	4	4
Total	9	12	9
Mean age	18.75 ± 9.06	55.2 ± 14.13	

## Data Availability

All data created or analyzed during this study are available from the corresponding author upon request.

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
