# Peer review of "Plasma Somatostatin Levels Increase during Scoliosis Surgery, but Not Herniated Disc Operations: Results of a Pilot Study"

_biomedicines, 2023, doi:10.3390/biomedicines11082154_

Round 1
Reviewer 1 Report
With relatively small number of patients (30), i would add in Title: Preliminary results. This marginal number of patients needs to be stressed in study limitation section, which should be added in discussion.
At the end, conclusion section should be added with succint conclusion points.
Overall, nice study, recommend revision.
Minor English editing.
Reviewer 2 Report
Dear Authors,
your paper seems really interesting since it deals with the plasma somatostatin levels related to scoliosis surgery.
Some aspects have to be clarified.
Please check that the affiliations contain all the information required by the Journal editorial rules.
Then, it is not clear which is the study design. If you made a comparison between different surgery types, how did you establish the sample size of each group? And then, since you deepen pain-related aspects, how did you consider this aspect in the enrollment criteria? Did you collect data about the analgesic drugs that patients assumed before/after the surgery procedures? Please clarify these aspects.
Discussion seems still poor, You should improve this section.
You should also report the limitations of this study and you should create a separate paragraph for the conclusion at the end of the article.
Best regards and good luck!
Round 2
Reviewer 1 Report
na